# Shaking Table Testing and Numerical Study on Aseismic Measures of Twin-Tube Tunnel Crossing Fault Zone with Extra-Large Section

Fengbing Zhao [1,*] 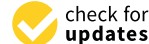, Bo Liang [1], Ningyu Zhao [1,2] and Bolin Jiang [3]

1   School of Civil Engineering, Chongqing Jiaotong University, No. 66, Xuefu Road, Nan'an District, Chongqing 400074, China; liang_laoshi@126.com (B.L.); zny2008@163.com (N.Z.)
2   State Key Laboratory of Mountain Bridge and Tunnel Engineering, No. 66, Xuefu Road, Nan'an District, Chongqing 400074, China
3   Chongqing Vocational Institute of Engineering, Chongqing 402260, China; bolinjiang@cqvie.edu.cn
*   Correspondence: zhao.fengbing@163.com

**Abstract:** As transportation networks continue to expand into mountainous regions with high seismic activity, ensuring the seismic safety of tunnels crossing active faults has become increasingly crucial. This study aimed to enhance our understanding of the impact of fault zones on the seismic behavior of tunnels and to provide optimized seismic design recommendations through a comprehensive experimental and numerical investigation. The focus of this research is the Xiangyangshan Highway Tunnel in China, which intersects a significant longitudinal fault. Large-scale shake table tests were performed on 1:100 scale physical models of the tunnel to analyze the seismic responses under various ground motion excitations. Detailed three-dimensional finite difference models were developed in FLAC3D and calibrated based on the shake table results. The tests indicated that strains, earth pressures, and accelerations experience localized amplification within 10–20 m of the fault interface compared to undisturbed ground sections. Common seismic mitigation measures, such as rock grouting, seismic joints, and shock absorption layers, were observed to effectively reduce the amplified seismic demands. Grouting, in particular, led to an average reduction of up to 56.3% in circumferential strain and 38.5% in earth pressure. It was concluded that 6 m thick grouted zones and 20 cm thick rubber interlayers between tunnel lining shells provide optimal structural reinforcement against the effects of fault zones. This study provides valuable insights for improving the seismic resilience of underground transportation corridors in seismically active regions.

**Keywords:** extra-large section tunnel; shaking table test; surrounding rock grouting; shock absorption layer; seismic joint; fault

## 1. Introduction

With rapid urbanization and economic development, the construction of transportation infrastructure has significantly accelerated throughout China in recent decades. The expansion of highways, high-speed railways, and underground metro networks has increasingly focused on effectively traversing complex terrain, particularly in mountainous regions [1–3]. Consequently, tunnels have become crucial for facilitating continuous travel through areas with challenging geological conditions. However, tunnels that cross active faults or are situated in high seismic zones are especially susceptible to earthquake damage, which can jeopardize structural integrity and public safety [4]. As a result, the optimized seismic design of tunnels is of utmost importance from both engineering and societal standpoints.

Numerous seismic zones with frequent seismic activity are located in Southwest and Northwest China. The construction of large-section tunnels in areas affected by active faults and high-intensity seismic zones presents an unavoidable challenge. In accordance with

the Specification for Seismic Design of Highway Tunnels (JTG 2232-2019) [5], the Standard for Seismic Design of Underground Structures (GB/T 51336-2018) [6], the Code for Seismic Design of Railway Engineering (GB 50111-2006) [7], and other standards, measures such as surrounding rock reinforcement, seismic joints, lining reinforcement, and extension of open spaces can be implemented for tunnel linings in high-intensity areas. While engineers and scholars have studied and compiled some effective vibration control measures for these tunnels with complex adverse site conditions, further research is still needed to investigate the vibration control mechanisms and key parameters of these measures [1].

In high-intensity seismic areas, shallow tunnel structures are more susceptible to severe deformation or even damage. Moreover, in mountainous regions, the complex geological conditions exacerbate the degree of structural damage [8]. Currently, scholars have developed mature theories on earthquake damage types, dynamic response characteristics, and laws governing ordinary mountain tunnels [5,6]. Previous studies have also demonstrated that shallow tunnels located within or near fault zones experience more severe deformations and damages compared to those in homogeneous ground. However, there is relatively limited research on the seismic dynamic response laws of shallow buried sections of extra-large highway tunnels.

The common anti-vibration measures outlined in the aforementioned Chinese standards include surrounding rock reinforcement methods, such as grouting reinforcement and anchor reinforcement, as well as lining reinforcement methods like concrete strength improvement, lining thickening, and steel mesh lining. In 2016, Xu et al. [9] conducted shaking table tests on these measures, analyzing their anti-vibration effects and studying the action mechanism and impact of seismic measures for mountain tunnels. While such measures have demonstrated effectiveness through small-scale shaking table tests, their vibration control mechanisms and key parameters require further elucidation, especially for tunnels with deep burial in complex fault settings.

Numerical dynamic analysis and shaking table tests are commonly employed to investigate anti-vibration measures for tunnels. The numerical dynamic time-history analysis method of the full model can comprehensively describe the dynamic interaction of surrounding rock and the tunnel, fully accounting for the geometric irregularity and medium inhomogeneity of the site [10]. This method is the most practical approach for predicting the effects of various anti-vibration measures prior to their implementation in tunnels and underground structures. Shaking table test methods are divided into centrifuge shaking table tests and ordinary shaking table tests, each with the characteristic of strictly controlling the research parameters of the test object and being unaffected by external environmental and natural conditions [11]. General shaking table test equipment is typically large and can conduct large-scale model tests, with relatively minimal model size effects on the test. In contrast, the model size in centrifuge tests is generally small, and the test results are easily influenced by boundary effects [12]. Given the characteristics of these research methods, the numerical dynamic analysis method is generally utilized to analyze the dynamic response characteristics of the tunnel structure, and subsequently, the shaking table test is employed to validate the dynamic response characteristics, yielding significant effects.

Common numerical analysis methods include the finite element method, finite difference method, discrete element method, and others. The finite element method is commonly implemented using software such as ANSYS 10.0 and ABAQUS 6.11, while the finite difference method typically utilizes software like FLAC3D 9.0. Xie [13], in 2013, employed the finite element analysis method to compare and evaluate the seismic behavior of a tunnel with different cross-sectional shapes, focusing on a portal segment in a shallow-built tunnel located in a seismic region with strong motion. Similarly, Salemi et al. [14], in 2018, investigated the behavior of the concrete lining of circular shallow tunnels in sedimentary urban areas under seismic loads using an integration of numerical and metaheuristic techniques. Furthermore, Momenzadeh et al. [15], in 2019, explored the function of the tunnel lining under static and seismic conditions by combining the response surface method, the Hasofer-

Lind reliability concept, and the finite element method, with a focus on the reliability of the lining system of a small underground tunnel in the soil. An et al. [16], in 2021, established an ABAQUS finite element model to clarify the influence of fiber-reinforced concrete lining structure on the seismic performance of an urban shallow-buried rectangular tunnel. Additionally, Jian et al. [17], in 2022, applied the finite element method to examine the influence of the thickness of a shock absorption layer on the seismic effect of an urban shallow-buried double-arch rectangular tunnel. Moreover, Liu et al. [18], in 2022, employed full dynamic time history analysis to investigate the interaction in the transversal direction of an arched tunnel buried in a stratified soil in Hohhot, China. A numerical parametric analysis was conducted to elucidate critical response characteristics.

Shaking table tests for tunnels are used to validate numerical calculation results or theoretical analysis findings, and to independently analyze the dynamic response characteristics of tunnel structures. The comparative analysis of several anti-vibration measures by Xu et al. [9] falls into the latter category. Shaking table model tests were conducted to investigate the seismic behaviors of a double box utility tunnel with joint connections and the surrounding soil [19]. Liang et al. [20] investigated the seismic behavior of a shield tunnel with an ultra-large diameter of 15 m passing through a soft-hard stratum using a series of 1/30 scaled shaking table model tests and numerical simulations. Additionally, in 2022, Zhang et al. [21] studied the seismic performance of a shield tunnel under near-field ground motion, conducting a series of large-scale shaking table tests.

This study aimed to enhance the understanding of seismic performance for an extra-large cross-section highway tunnel crossing faults through a comprehensive experimental and numerical investigation. A representative case tunnel in Southwest China was selected as the research object. Shaking table tests were initially conducted on physical models to reveal strain, earth pressure, and acceleration distributions with and without control measures, including surrounding rock grouting, seismic joints, and shock absorption layers. Based on the test results, finite difference analyses were then performed to optimize key parameters of the grouting thickness and the shock absorption layer thickness. The findings provide practical seismic design recommendations applicable to similar tunnels constructed in complex fault environments.

## 2. Case Study Tunnel

The case study tunnel selected for this research is the Xiangyangshan Tunnel located along the Qujing-Kunming Expressway in Yunnan Province, Southwest China. The Xiangyangshan tunnel is situated in the middle of the eastern Yunnan platform fold belt. The geological structure is highly intricate, particularly with a well-developed fault structure. The clearance width of a single tunnel on the standard section is 15.53 m, and the maximum excavation area is 167.6 m$^2$. It is a three-lane extra-large section highway tunnel. The tunnel is designed as a left and right tunnel with a typical shallow buried section and a double tunnel spacing of 23.07 m. The cross-sectional dimensions of the lining structure and the tunnel's portal site under construction are illustrated in Figure 1. The surrounding rock mass of the tunnel is categorized as grades $V_2$ to $III_1$, as assessed in accordance with the Chinese industry standard "Code for Design of Road Tunnel (JTG D70-2004)" [22]. The tunnel exhibits numerous fault fracture zones, developed joints, and fissures, leading to a rock mass that is relatively fragmented and unstable.

Based on the geological survey report, three low-resistance fractures have been identified in the surrounding rock of the tunnel, suspected to be structural fracture zones. The shallow rock in the tunnel area exhibits significant weathering, resulting in poor rock integrity. The deep joint fissures range from underdeveloped to moderately developed, with the surface primarily featuring two sets of joint fissures, which are detrimental to the stability of the surrounding rock of the tunnel.

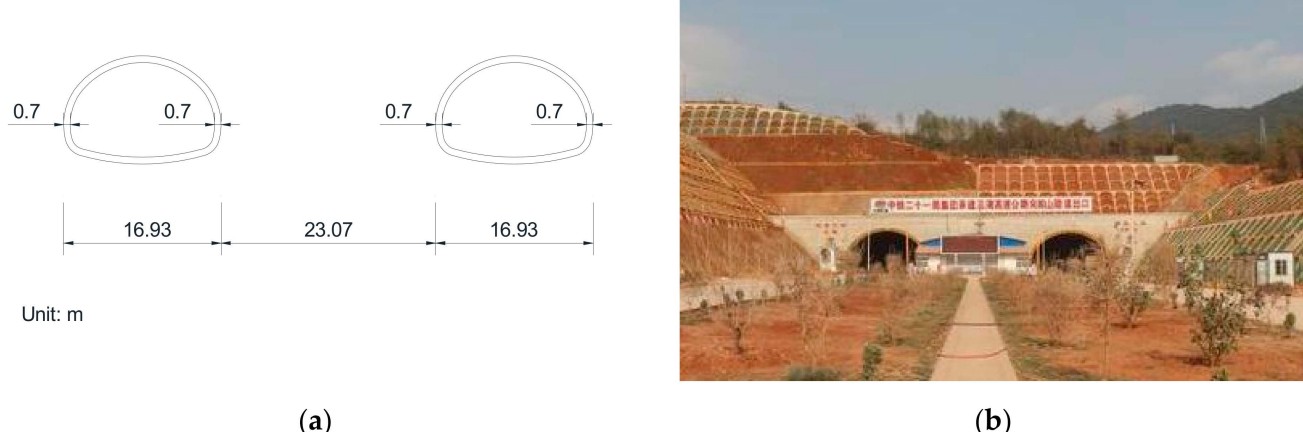

**Figure 1.** Tunnels cross section parameters and its portal site under construction: (**a**) cross section parameters; (**b**) the portal site under construction.

The basic seismic intensity of the tunnel site area is rated at VIII on China's intensity scale. However, to account for the critical function of the expressway, a design fortification intensity of IX degree (strong earthquake level) was adopted for seismic design and control measures. The peak ground acceleration value for tunnel analysis was conservatively set at 0.4 g. Three types of input earthquake waves were incorporated into the subsequent physical model tests, namely the El-Centro, Taft, and artificial synthetic waves. The former two consist of seismic wave records from renowned historical earthquakes, whereas the latter are artificial seismic waves generated using probability function techniques. These waves represent different seismic characteristics that the tunnel may encounter. The peak value of original records was also adjusted and used in these physical model tests when input.

## 3. Shake Table Model Tests

To experimentally investigate the seismic responses of the Xiangyangshan Tunnel and evaluate potential vibration control measures, physical model tests were conducted on a large-scale shake table. This section describes the testing methodology and presents the results obtained, which provide valuable data for calibrating numerical models and optimizing anti-seismic designs.

### 3.1. Model Design and Fabrication

Given the intricate and time-consuming nature of the model testing process, as well as the extended duration required for each experiment, only two tunnel models were constructed. Considering the research scope and testing conditions, the geometric similarity ratio of the test model is set at 1/100. The geotechnical model is designed to have dimensions of 91 cm in total height, 200 cm in total length, and 137 cm in total width, with a mass density of 1. The elastic modulus similarity ratio is 1/100, and other physical quantities are scaled accordingly based on the principles of similarity. The physical parameters and similarity relationships between the model and prototype are outlined in Table 1. Table 2 provides detailed information on the setup conditions of the two test models. Figures 2 and 3 show the size and preparation of the shake table model.

**Table 1.** Physical parameters and similarity relationships between experimental models and prototypes.

| Physical Parameters | Similarity Relationships | Physical Parameters | Similarity Relationships |
|---|---|---|---|
| Stress $\sigma$ | 1/137.5 | Time $T$ | 1/11.72604 |
| Strain $\varepsilon$ | 1/1.375 | Frequency $f$ | 1/0.08528 |
| Elastic modulus $E$ | 1/100 | Velocity $v$ | 1/11.72604 |
| Poisson's ratio $\mu$ | 1 | Accelerate $a$ | 1/0.727263 |
| Density $\rho$ | 1/1.375 | gravitational acceleration g | 1 |
| Length $L$ | 1/100 | Cohesive force c | 1/137.5 |
| Displacement $u$ | 1/137.5 | Damping C | 1/1612.33 |
| Force $F$ | 1/1375000 | Stiffness K | 1/10000 |

**Table 2.** Model setup details.

| Model | Setting Conditions | Proposed Research Content | Model Overall Size |
|---|---|---|---|
| Model 1 | The fault has a dip of 90 degrees and a width of 20 m, and the tunnel is a two-hole tunnel with a small clearance distance. The left tunnel does not have any anti-vibration measures and is supported by general V surrounding rock conditions. In contrast, grouting is used to reinforce the surrounding rock in the right tunnel. The grouting is carried out at intervals of the entire ring with a thickness of 4 m. Different reinforcement lengths are set before and after the longitudinal upper fault, with lengths of 25 m and 35 m, respectively. | (1) Seismic response characteristics of an extra-large section tunnel. (2) Strengthening effect of grouting reinforcement on surrounding rock in fault-related areas. | 137 cm × 200 cm × 91 cm based on similarity ratio calculation. |
| Model 2 | The fault has a dip of 90 degrees and a width of 20 m, and the tunnel is a two-hole tunnel with a small clearance distance. Both the left and right tunnels are supported by Class V surrounding rock. The left tunnel is equipped with seismic joints, while the right tunnel is provided with shock absorption layers. Different reinforcement ranges are set before and after the longitudinal upper fault, with lengths of 25 m and 35 m, respectively. | (1) Establishing the strengthening effect of seismic joints. (2) Establishing the strengthening effect of the shock absorption layer. | |

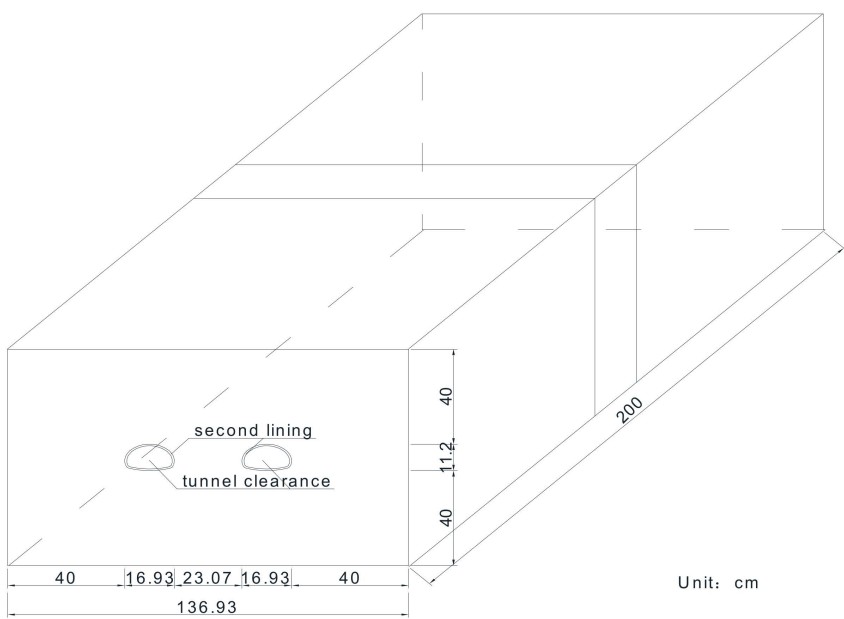

**Figure 2.** Size of the shaking table model.

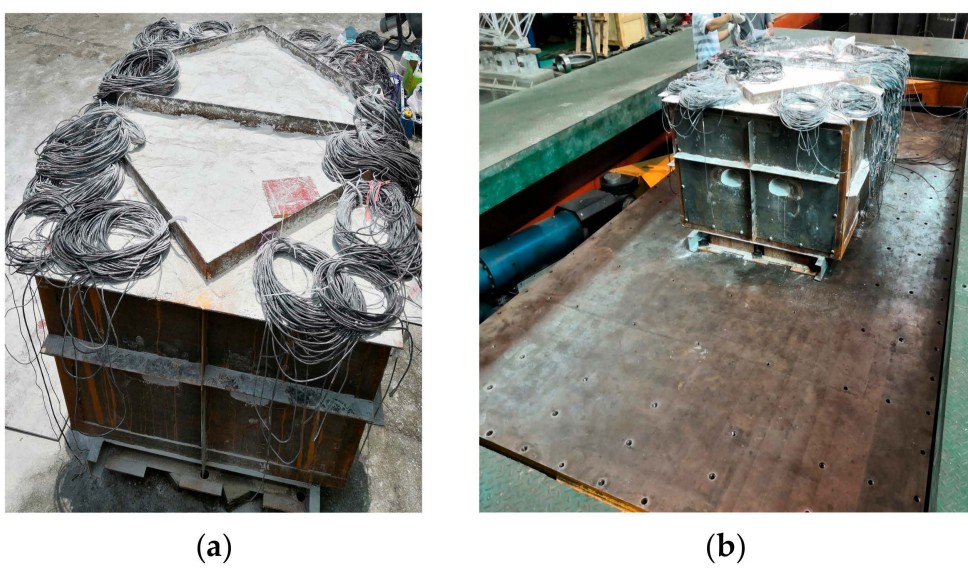

**Figure 3.** Descriptions of (**a**) model completion and (**b**) mounting on shaking table.

### 3.2. Instrumentation and Measurement

Strain sensors were positioned at designed monitoring sections A–A, B–B, C–C, D–D, E–E, F–F, G–G, H–H, I–I, J–J, and K–K (refer to Figure 4). A total of six strain gauge measuring points were placed on the outer surface of a single tunnel lining (see Figure 5). Each measuring point was equipped with two longitudinal and circumferential test channels, denoted by the symbol "Y" for strain measurement. Dynamic earth pressure testers and accelerometers were installed in sections C–C, D–D, E–E, F–F, G–G, H–H, and I–I. A dynamic earth pressure gauge (marked by the symbol "○" and sequentially labeled with number after the letter P in Figure 6) was positioned near the middle of each tunnel vault and left wall, respectively. An accelerometer was arranged at the top of the model box tunnel for each X/Y direction, and another accelerometer was placed at the bottom plate of the model box for each X/Y/Z direction. Additionally, two accelerometers were positioned at the middle of the right wall and the vault, one for each X/Y/Z direction.

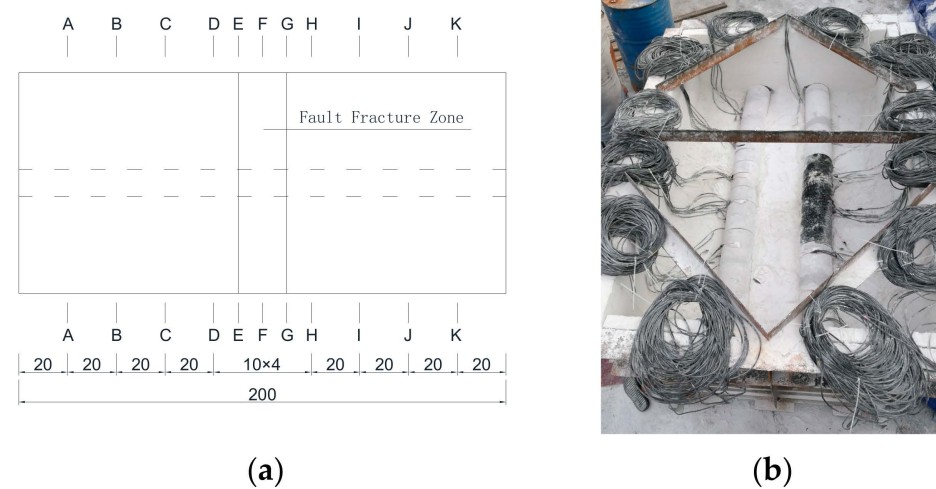

(**a**)                                                                                   (**b**)

**Figure 4.** Diagram of (**a**) longitudinal numbering of monitoring sections and (**b**) sensor arrangement.

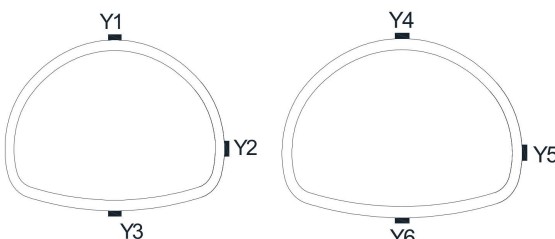

**Figure 5.** Arrangement of strain gauges within a single tunnel section.

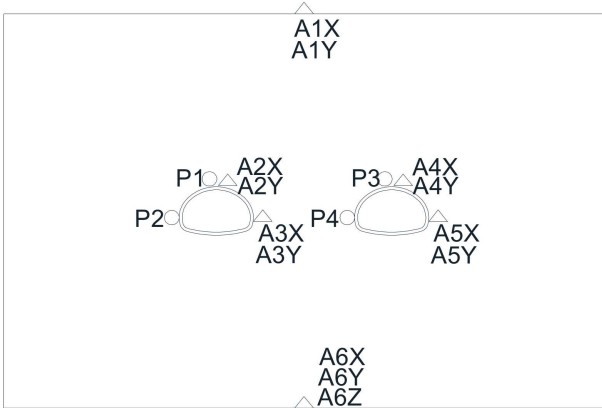

**Figure 6.** Layout of accelerometers and earth pressure gauges in the model cross-section.

### 3.3. Loading Protocols

The experimental scenarios involved exposing models to a range of input ground motions derived from scaled prototype earthquake data. The testing parameters for both models remained uniform, as outlined in Table 3. Four distinct seismic input levels were employed, specifically 0.25 A, 0.50 A, 0.75 A, and 1 A, with "A" denoting the peak value of the design seismic wave. Consistent peak amplitudes were maintained for identical El Centro, Taft, and artificial waves at each intensity level to facilitate the comparative analysis of structural reactions across various ground motion attributes.

**Table 3.** Model test conditions.

| No. | Loading Model | Input Seismic Wave | Seismic Input Mode | Model Loading Intensity |
|---|---|---|---|---|
| 1 | Model 1 and Model 2 | White noise | Bottom synchronous input X, Y, Z direction vibration | Load seismic wave peak 0.10 A |
| 2 | Model 1 and Model 2 | El-Centro wave | Bottom input X direction vibration | Load seismic wave peak 0.25 A |
| 3 | Model 1 and Model 2 | Taft wave | Bottom input X direction vibration | Load seismic wave peak 0.25 A |
| 4 | Model 1 and Model 2 | Artificial wave | Bottom input X direction vibration | Load seismic wave peak 0.25 A |
| 5 | Model 1 and Model 2 | White noise | Bottom synchronous input X, Y, Z direction vibration | Load seismic wave peak 0.10 A |
| 6 | Model 1 and Model 2 | El-Centro wave | Bottom input X direction vibration | Load seismic wave peak 0.5 A |
| 7 | Model 1 and Model 2 | Taft wave | Bottom input X direction vibration | Load seismic wave peak 0.5 A |
| 8 | Model 1 and Model 2 | Artificial wave | Bottom input X vibration | Load seismic wave peak 0.5 A |
| 9 | Model 1 and Model 2 | White noise | Bottom synchronous input X, Y, Z direction vibration | Load seismic wave peak 0.10 A |
| 10 | Model 1 and Model 2 | El-Centro wave | Bottom input X direction vibration | Load seismic wave peak 0.75 A |
| 11 | Model 1 and Model 2 | Taft wave | Bottom input X direction vibration | Load seismic wave peak 0.75 A |
| 12 | Model 1 and Model 2 | Artificial wave | Bottom input X direction vibration | Load seismic wave peak 0.75 A |
| 13 | Model 1 and Model 2 | White noise | Bottom synchronous input X, Y, Z direction vibration | Load seismic wave peak 0.10 A |
| 14 | Model 1 and Model 2 | El-Centro wave | Bottom input X direction vibration | Load seismic wave peak 1 A |
| 15 | Model 1 and Model 2 | Taft wave | Bottom input X direction vibration | Load seismic wave peak 1 A |
| 16 | Model 1 and Model 2 | Artificial wave | Bottom input X direction vibration | Load seismic wave peak 1 A |

Before excitation, a low-amplitude white noise signal was utilized to consolidate the soil model, and white noise scanning was introduced each time the peak acceleration value was adjusted to observe the dynamic behavior of the system model. White noise with a maximum amplitude of 0.1 g was applied in scenarios 1, 11, 21, and 31 to eliminate initial residual deformations and other variables. The original design seismic wave acceleration peak value was 0.4 g, equivalent to $4.0 \, \text{m/s}^2$, with a vibration duration of 40.00 s. Following adjustments based on the similarity principle, the acceleration peak value was revised to $5.5 \, \text{m/s}^2$, and the vibration duration was set at 3.4112 s. Consequently, the maximum peak value "A" of the seismic wave was established at $5.5 \, \text{m/s}^2$.

*3.4. Results and Analysis*

3.4.1. Strain Analysis

The assessment of circumferential and longitudinal strain along the tunnel's axial direction under different seismic wave excitations and anti-vibration strategies was carried out using the findings from the model tests, illustrated in Figures 7–9. The testing segment of the tunnel model is positioned within the vertical range of 90–110 cm.

In general, the circumferential and longitudinal strains within and around the fault section (90–110 cm) displayed slightly higher values compared to other undisturbed ground segments when they were subjected to the three input ground motions. This indicated

that the boundary effects of the fault zone reached a certain distance into the surrounding intact rock.

After incorporating anti-vibration measures, the strain distributions exhibited a more evenly spread pattern along the tunnel axis in contrast to the initial condition without any interventions. Additionally, the peak strain values decreased with the implementation of these controls. Although certain local strain readings displayed slight increments, these variations were presumed to have stemmed from minor testing inaccuracies.

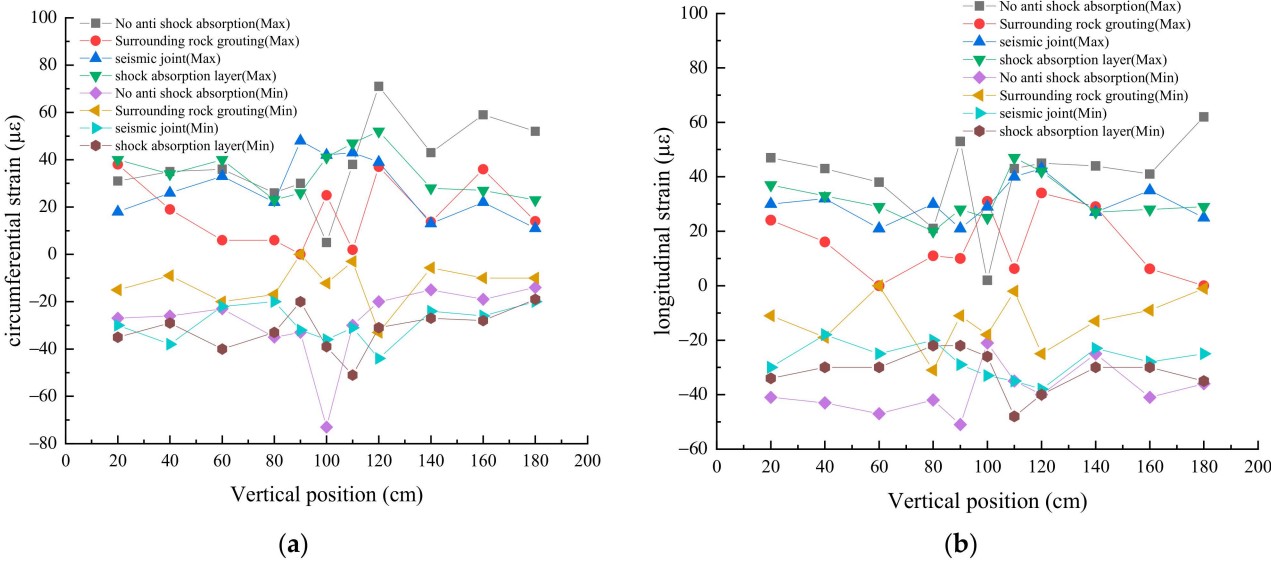

(**a**)  (**b**)

**Figure 7.** Peak distribution of (**a**) circumferential and (**b**) longitudinal strain increments along the longitudinal direction of the tunnel under the El-Centro wave.

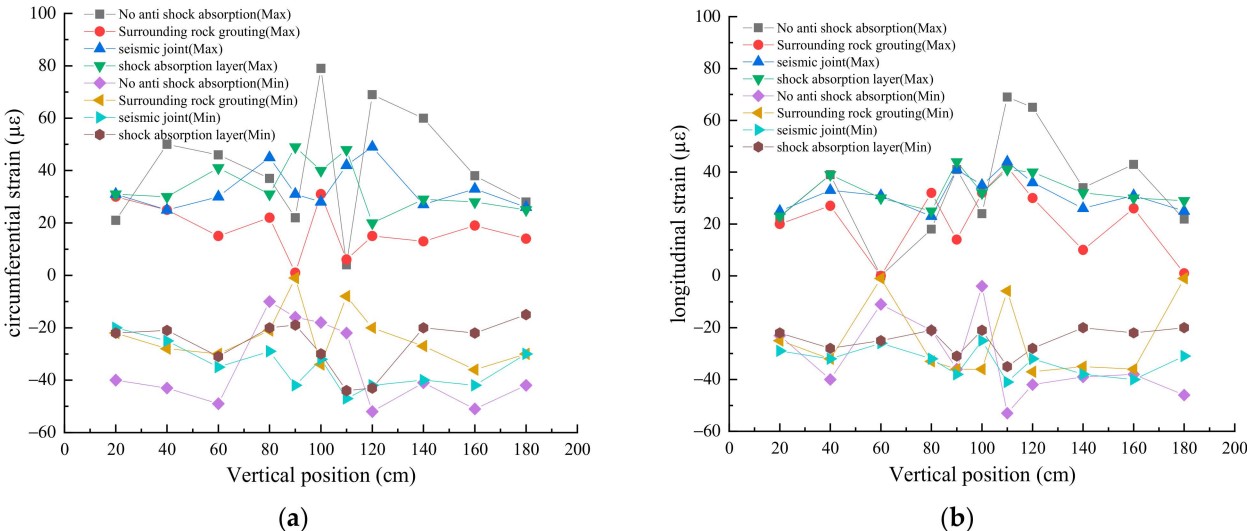

(**a**)  (**b**)

**Figure 8.** Peak distribution of (**a**) circumferential and (**b**) longitudinal strain increments along the longitudinal direction of the tunnel under the Taft wave.

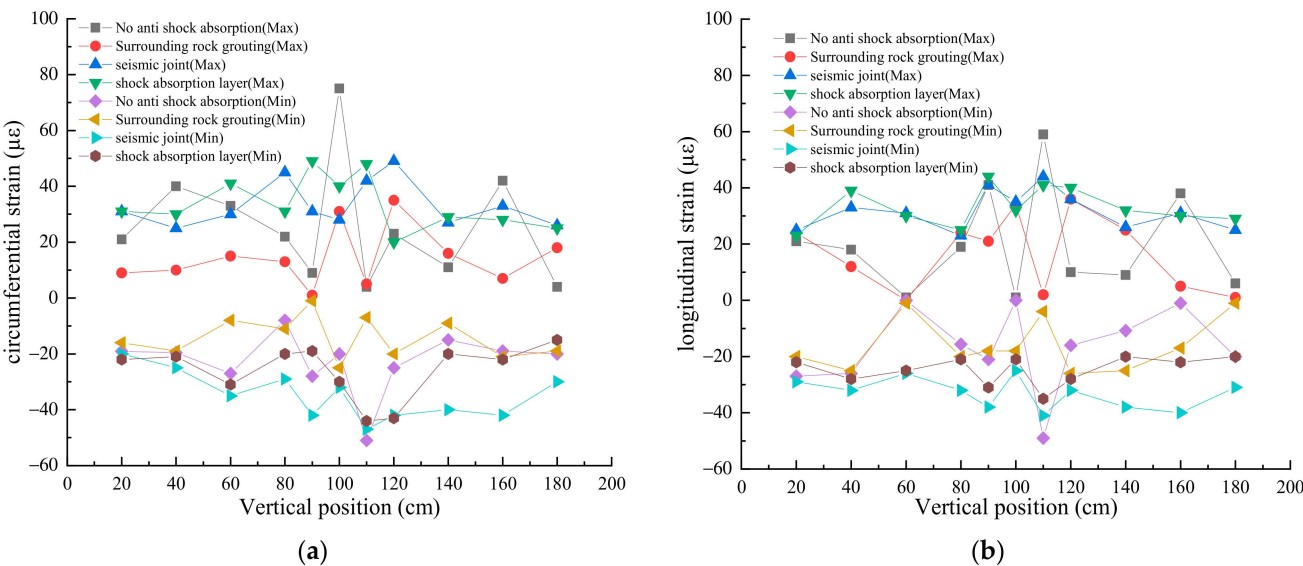

**Figure 9.** Peak distribution of (**a**) circumferential and (**b**) longitudinal strain increments along the longitudinal direction of the tunnel under the artificial wave.

In particular, when subjected to El-Centro wave excitation (refer to Figure 7), the introduction of anti-seismic measures effectively mitigated strain responses. In the absence of these measures, the maximum and minimum circumferential strain values were recorded at 71 με and −73 με, respectively. Subsequent to the implementation of interventions such as surrounding rock grouting, seismic joints, and shock absorption layers, reductions in strain levels were evident. Specifically, the maximum circumferential strain decreased to 38 με, 48 με, and 52 με, corresponding to the reduction ratios of −46.5%, −32.4%, and −26.8%, respectively. Similarly, the minimum circumferential strain values decreased to −33 με, −38 με, and −51 με, with reduction ratios of −54.8%, −47.9%, and −30.1%. As for longitudinal strain, the initial maximum value in the absence of measures was recorded at 62 με. Post-grouting, seismic joints installation, and shock absorption layer application, this figure decreased to 34 με, 43 με, and 47 με, respectively. The reduction ratios for longitudinal strain were calculated at −45%, −30.64%, and −24.2%.

Under Taft wave excitation, as illustrated in Figure 8, the implementation of surrounding rock grouting, seismic joints, and shock absorption layers effectively reduced strains. The reduction ratios of maximum circumferential strain were −60.8%, −38%, and −38%, respectively, for each measure. Meanwhile, the reduction ratios of minimum circumferential strain were −33.3%, −7.8%, and −13.7%, respectively. In terms of longitudinal strain, the reduction ratios of maximum strain were −53.6%, −36.2%, and −36.2% following the application of the three controls. The reduction ratios of minimum longitudinal strain were −30.2%, −22.6%, and −34%, respectively.

Similar trends were observed under artificial wave excitation, as depicted in Figure 9. In this case, the reduction ratios of maximum circumferential strain were −53.3%, −34.7%, and −34.7% with each control. The reduction ratios of minimum circumferential strain were −51.9%, −9%, and −14.8%, respectively. For longitudinal strain, the reduction ratios of maximum strain ranged from −39% to −25.4%, while the reduction ratios of minimum strain ranged from −47% to −16.3%.

Furthermore, the analysis indicated a shift in the locations of peak strain at the fault interface section after the installation of seismic joints and shock absorption layers, demonstrating the effectiveness of these measures against fault zone amplification effects.

### 3.4.2. Earth Pressure Analysis

Figure 10 illustrates the peak distribution of lining earth pressure increments along the tunnel longitudinal direction under different anti-vibration measures implemented in the

model tests. Generally, the earth pressure increments near the fault section were slightly higher than in other intact sections under all three input ground motions. The distribution became more uniform after the application of anti-seismic controls.

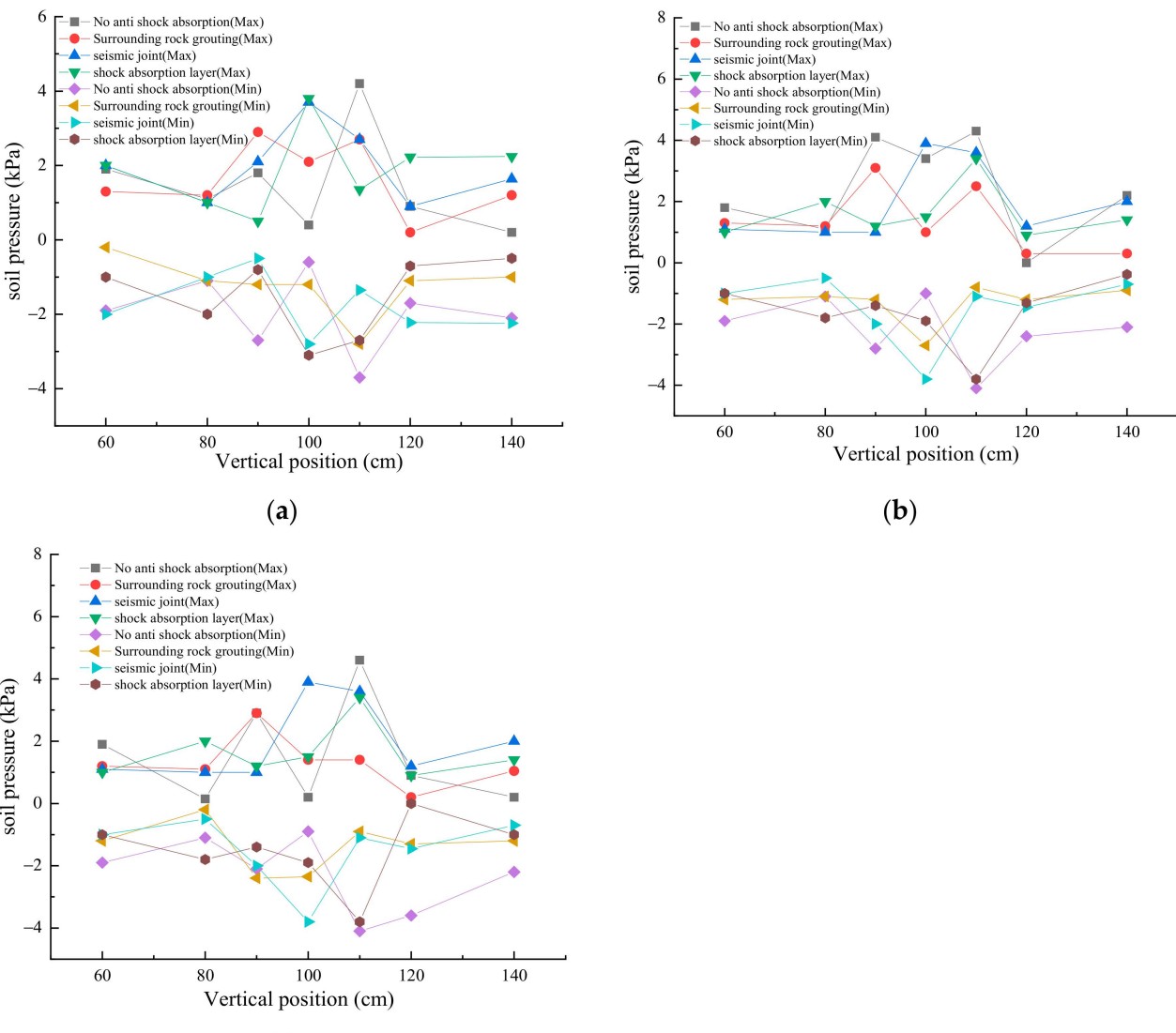

**Figure 10.** Increment peak distribution of lining earth pressure along the longitudinal direction of the tunnel: (**a**) El-Centro wave; (**b**) Taft wave; (**c**) artificial wave.

In Figure 10a, it was evident that under El-Centro wave excitation, the maximum and minimum increments of lining earth pressure were 4.2 kPa and −3.7 kPa, respectively, without seismic measures. After grouting, installing seismic joints, and incorporating a shock absorption layer, the maximum increment of lining earth pressure decreased to 2.9 kPa, 3.7 kPa, and 3.8 kPa, respectively, with reduction ratios of −40%, −12%, and −9.5%. Simultaneously, the minimum increment of lining earth pressure was −2.8 kPa, −2.8 kPa, and −3.1 kPa, respectively, with reduction ratios of −24.3%, −24.3%, and −16.2%.

In Figure 10b, under Taft wave excitation, following the grouting of surrounding rock, installation of anti-seismic joints, and implementation of shock absorption layers, the reduction ratio for the maximum increment of lining earth pressure was −28%, −9.3%, and −20.9%, respectively. Similarly, the reduction ratio for the minimum increment of lining earth pressure was −34%, −7.3%, and −7.3%, respectively. In Figure 10c, under artificial wave excitation, after the completion of grouting, installation of anti-seismic measures,

and integration of shock absorption layers, the reduction ratio for the maximum increment of lining earth pressure was −37%, −15.2%, and −26.1%, respectively. Additionally, the reduction ratio for the minimum increment of lining earth pressure was −41.4%, −7.3%, and −7.3%, respectively.

Furthermore, the analysis of the peak position of earth pressure indicated that the peak position of earth pressure on the fault interface section changed after the installation of seismic joints and shock absorption layers.

### 3.4.3. Acceleration Analysis

Figures 11–13 illustrate the distribution of peak acceleration values in the X and Y axis along the tunnel section under different anti-vibration measures implemented in the model tests. The results revealed dynamic shifts in the peak positions of X-axis and Y-axis acceleration within the tunnel section when subjected to three different types of wave excitations, in contrast to scenarios lacking anti-vibration interventions. With the implementation of anti-vibration measures, the distribution of acceleration increments in the tunnel section exhibited a more uniform pattern, thus successfully managing the acceleration peaks in both the X and Y axes.

In Figure 11, it was observed that under El-Centro wave excitation, the maximum and minimum values of acceleration in the X axis were 7.2 m/s$^2$ and −7.1 m/s$^2$, respectively, without seismic measures. After grouting, installing anti-seismic joints, and incorporating a shock absorption layer, the maximum value of acceleration in the X axis decreased to 5.2 m/s$^2$, 5.2 m/s$^2$, and 6.3 m/s$^2$, respectively, with reduction ratios of −27.8%, −27.8%, and −12.5%, while the minimum value of acceleration in the X axis was −5.2 m/s$^2$, −5.2 m/s$^2$, and −6.5 m/s$^2$, respectively, with reduction ratios of −26.8%, −26.8%, and −8.5%. Without seismic measures, the maximum and minimum values of acceleration in the Y axis were 5.3 m/s$^2$ and −5.2 m/s$^2$, respectively. After grouting, setting seismic joints, and incorporating a shock absorption layer, the maximum acceleration in the Y axis decreased to 2.3 m/s$^2$, 4.8 m/s$^2$, and 4.8 m/s$^2$, respectively, with reduction ratios of −56.6%, −9.4%, and −9.4%, while the minimum acceleration in the Y axis was −2.8 m/s$^2$, −4.9 m/s$^2$, and −4.8 m/s$^2$, respectively, with reduction ratios of −46.2%, −7.55%, and −7.7%, respectively.

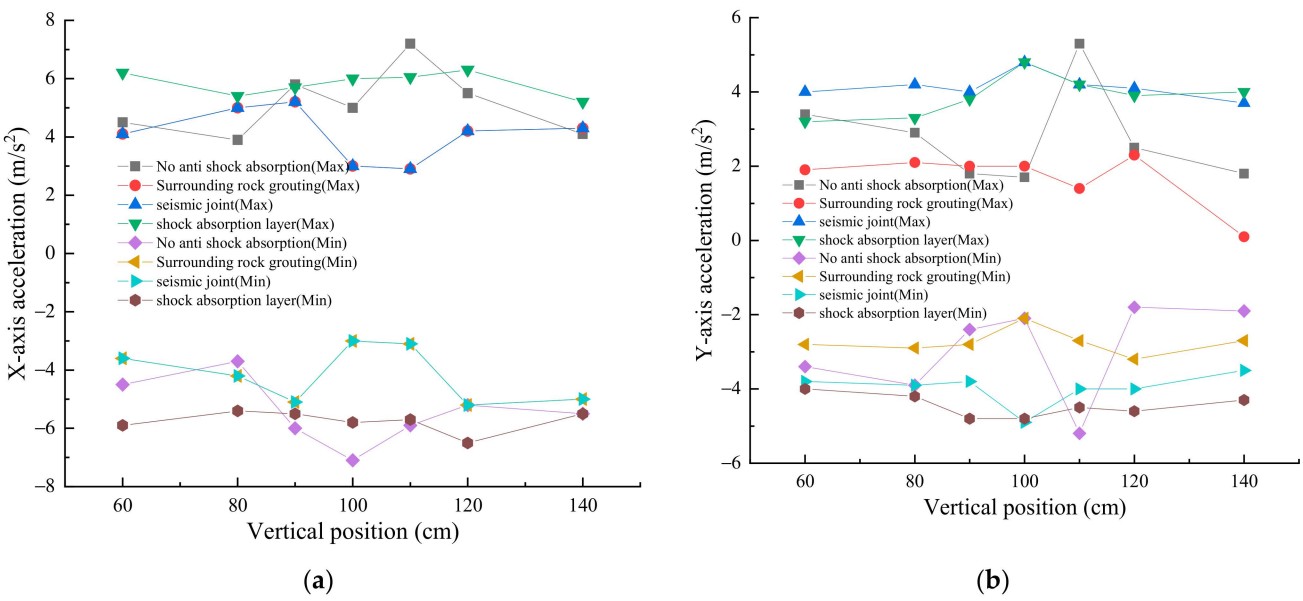

(**a**)  (**b**)

**Figure 11.** Distribution of (**a**) X-axis and (**b**) Y-axis acceleration peak value along tunnel longitudinal direction under El-Centro wave.

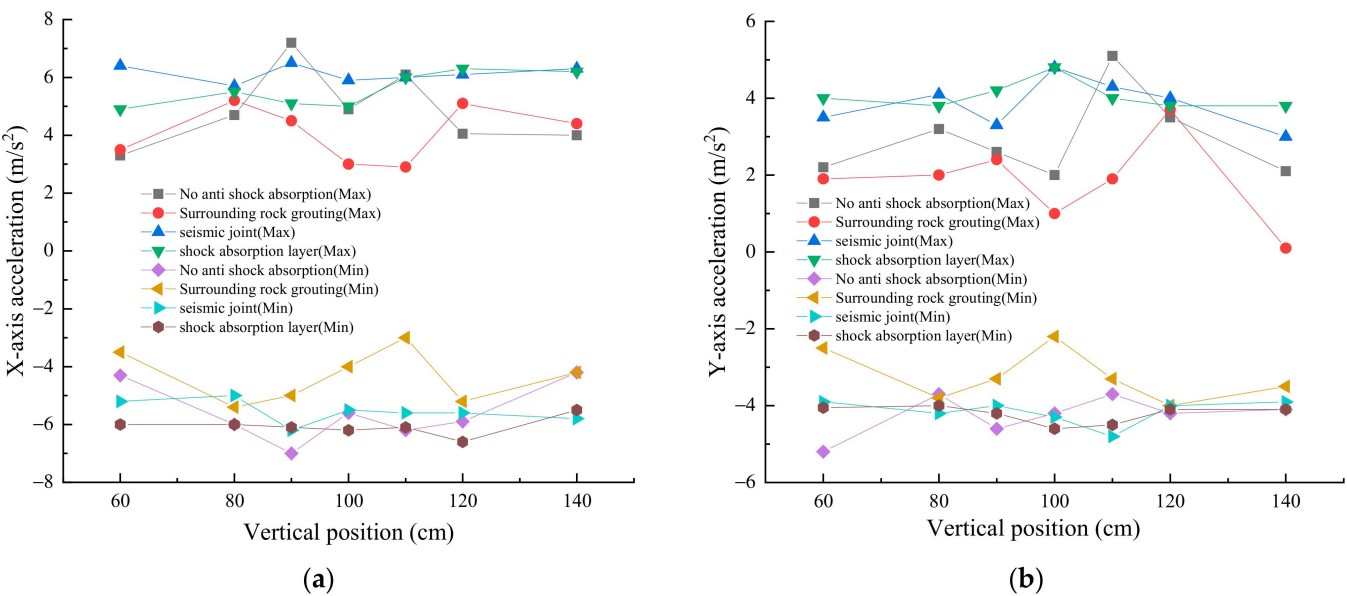

**Figure 12.** Distribution of (**a**) X-axis and (**b**) Y-axis acceleration peak value along tunnel longitudinal direction under Taft wave.

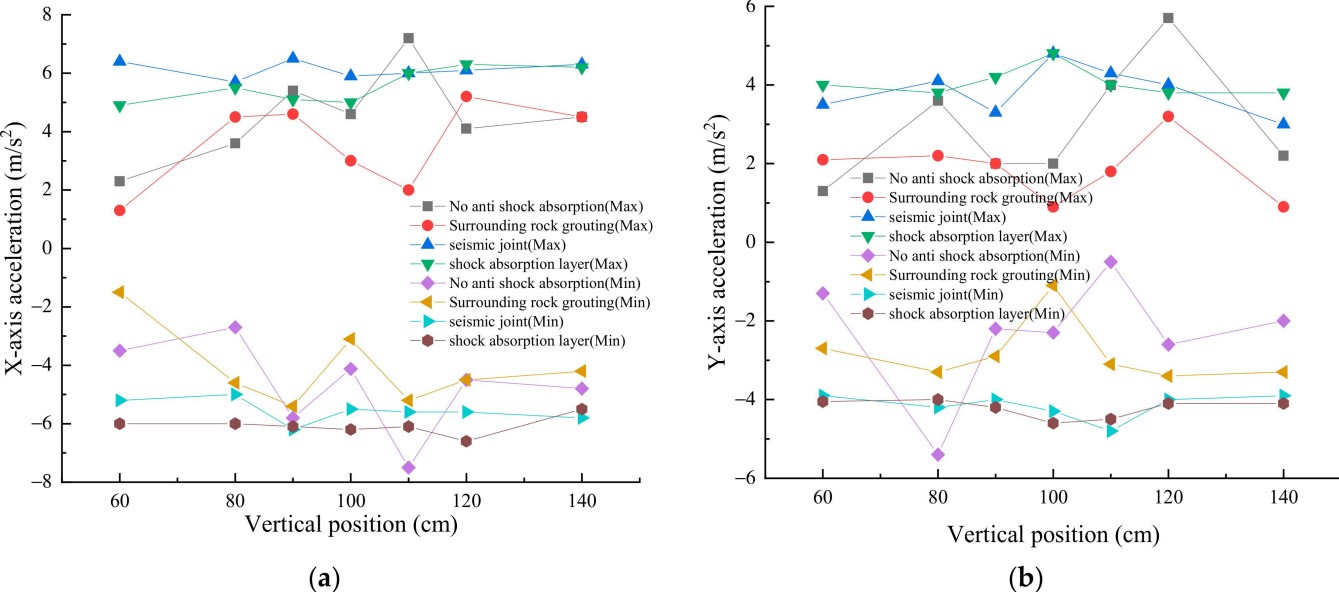

**Figure 13.** Distribution of (**a**) X-axis and (**b**) Y-axis acceleration peak value along tunnel longitudinal direction under artificial wave.

From Figure 12, it was evident that under Taft wave excitation, after grouting of surrounding rock, setting of seismic joints, and incorporating a shock absorption layer, the reduction ratio of the maximum acceleration in the X axis was −27.8%, −9.7%, −12.5%, respectively, while the reduction ratio of the minimum acceleration in the X axis was −22.9%, −11.4%, −5.7%, respectively. The reduction ratio of the maximum acceleration in the Y axis was −27.5%, −5.9%, −5.9%, respectively, and the reduction ratio of the minimum acceleration in the Y axis was −27%, −7.7%, −11.5%, respectively.

Likewise, from Figure 13, under artificial wave excitation, after grouting of surrounding rock, setting of seismic joints, and incorporating a shock absorption layer, the reduction ratio of the maximum acceleration in the X axis was −27.8%, −9.7%, −12.5%, respectively, while the reduction ratio of the minimum acceleration in the X axis was −28%, −17.3%, −12%, respectively. The reduction ratio of the maximum acceleration in the Y axis was

−43.8%, −15.8%, −15.8%, respectively, and the reduction ratio of the minimum acceleration in the Y axis was −37%, −11.1%, −14.8%, respectively.

Furthermore, the analysis of the peak position of acceleration indicated that the peak position of acceleration at the fault interface section changed after the installation of seismic joints and shock absorption layers.

### 3.4.4. Discussion

Based on the aforementioned test results, it was apparent that strain, earth pressure, and acceleration were slightly larger along the fault section and its vicinity under the excitation of three waveforms. In most cases, the index amplification effect was most pronounced at the interface of the fault zone surrounding rock. This demonstrated that the presence of a fault zone had a significant amplification effect on the acceleration, principal stress, and lining stress of the tunnel lining structure. The amplification effect extended a certain distance from the boundary surface of the surrounding rock of the fault zone to the complete surrounding rock section, with the maximum extension being 10 m. It was advisable to extend the minimum length of seismic fortification at both ends of the fault section by 10 to 20 m beyond the specifications outlined for tunnels with the recommended section in the Chinese industry standard (JTG D70-2004). Furthermore, after the installation of seismic joints and shock absorption layers, there were alterations in the positions of the maximum seismic response peaks at the fault interface sections.

To comprehensively analyze the shock absorption effect of surrounding rock grouting, seismic joints, and shock absorption layer, the maximum reduction ratio of strain, earth pressure, and acceleration peak values for each seismic measure was summarized in Table 4. In general, the trends in anti-vibration effects across various waveform excitations remained largely consistent, affirming the reliability of the model test outcomes.

**Table 4.** Analysis of anti-vibration effect.

| Anti-Shock Measures | Maximum Reduction Ratio (%) | | | | | |
|---|---|---|---|---|---|---|
| | Waveform | Circumferential Strain | Longitudinal Strain | Earth Pressure | X-Axis Acceleration | Y-Axis Acceleration |
| Surrounding rock grouting | El-Centro wave | −54.8% | −45% | −40% | −27.8% | −56.6% |
| | Taft wave | −60.8% | −53.6% | −34% | −27.8% | −27.5% |
| | Artificial wave | −53.3% | −47% | −41.4% | −28% | −43.8% |
| | Average | −56.3% | −48.5% | −38.5% | −27.9% | −42.6% |
| Seismic joints | El-Centro wave | −47.9% | −30.64% | −24.3% | −27.8% | −9.4% |
| | Taft wave | −38% | −36.2% | −9.3% | −11.4% | −7.7% |
| | Artificial wave | −34.7% | −30.5% | −15.2% | −17.3% | −15.8% |
| | Average | −40.2% | −32.4% | −16.3% | −18.8% | −11.0% |
| Shock absorbing layer | El-Centro wave | −30.1% | −24.2% | −16.2% | −12.5% | −9.4% |
| | Taft wave | −38% | −36.2 | −20.9% | −12.5% | −11.5% |
| | Artificial wave | −34.7% | −28.6 | −26.1% | −12.5% | −15.8% |
| | Average | −34.3% | −29.7% | −21.1% | −12.5% | −12.2% |

A comprehensive analysis revealed that the reinforcement of surrounding rock through grouting, the installation of seismic joints, and the addition of a shock absorption layer had noticeable seismic effects. Following the grouting reinforcement, the reduction ratios of hoop strain, longitudinal strain, earth pressure, peak acceleration value in the X direction, and peak acceleration value in the Y direction were −56.3%, −48.5%, −38.5%, −27.9%, and −42.6%, respectively. Subsequently, after the installation of seismic joints, the reduction ratios of hoop strain, longitudinal strain, earth pressure, X acceleration peak, and Y acceleration peak were −40.2%, −32.4%, −16.3%, −18.8%, and −11.0%, respectively. Finally, with the addition of the shock absorption layer, the reduction ratios of hoop strain, longitudinal strain, earth pressure, X acceleration peak, and Y acceleration peak were −34.3%, −29.7%,

−21.1%, −12.5%, and −12.2%, respectively. These results indicated that the seismic effects of seismic joints and shock absorption layers were similar, with the grouting reinforcement of surrounding rock exhibiting more significant effects than the former two.

## 4. Numerical Simulation

Following the validation achieved with the shake table experiments, extensive numerical analyses were conducted using the FLAC3D finite difference program. The objectives were to complement the physical model tests through more comprehensive parametric investigations and ultimately optimize key seismic control parameters for the design of Xiangyangshan Tunnel traversing the longitudinal fault zone.

### 4.1. Theoretical Basis

In numerical computations for models, the damping form typically adopts linear Rayleigh damping. In this case, the damping matrix [$C$] is represented as a linear combination of the mass matrix [$M$] and the stiffness matrix [$K$], as depicted in Equation (1) [23].

$$[C] = \alpha[M] + \beta[K], \tag{1}$$

where $\alpha$ and $\beta$ can be determined by assuming that the damping ratio for a specific second-order frequency is known.

The damping constant can be expressed using two frequencies of different magnitudes as follows:

$$\left. \begin{array}{l} \alpha = \frac{2\left(\xi_j\omega_i - \xi_i\omega_j\right)}{(\omega_i+\omega_j)(\omega_i-\omega_j)}\omega_i\omega_j \\ \beta = \frac{2\left(\xi_i\omega_i - \xi_j\omega_j\right)}{(\omega_i+\omega_j)(\omega_i-\omega_j)} \end{array} \right\}, \tag{2}$$

where $\xi_i$ and $\xi_j$ are the damping ratios of the $i$-th and $j$-th order circular frequencies ($\omega_i$ and $\omega_j$), respectively.

In general, engineering structures are usually assumed to have a damping ratio that remains relatively constant over a wide frequency range, with a typical value of 0.05 often used. The physical and mechanical parameters of the boundary elements in practical applications are determined by the materials of the adjacent surrounding rock medium, and specific calculation formulas can be found in the reference [24].

The input method for seismic waves acting on the computational model adopts the form of inputting stress waves at the viscoelastic artificial boundary, which is known as the equivalent boundary force method [25]. This method requires that the equivalent load applied to the artificial boundary makes the stress and displacement on the boundary identical to the original wave field. At this juncture, the equivalent boundary stress $F_B(t)$ can be expressed as follows:

$$F_B(t) = \sigma_B(t) + C\dot{u}_B(t) + Ku_B(t), \tag{3}$$

where $\sigma_B(t)$ represents the stress at the artificial boundary of the infinite domain model; $\dot{u}(t)$ and $u(t)$ represent the vibration velocity and displacement at the artificial boundary. $C$ and $K$, respectively, denote the damping coefficient and spring elasticity coefficient set at the artificial boundary.

When accounting for the fluctuation effects of the normal and tangential waves at the artificial boundary, researchers have analyzed the incident wave field of seismic waves in spherical coordinates and derived expressions for the vertical incident P-wave and S-wave at the artificial boundary, respectively [26].

Considering the incident non-attenuated S-wave in the far field, the corresponding expression is:

$$F_{BT}(t) = 2\rho C_S \dot{u}_B(t) + \frac{2G}{R} u_B(t),$$ (4)

where $R$ represents the distance from the artificial boundary to the ground or structure; $G$ represents the shear modulus of the adjacent medium in the artificial boundary region; $\lambda$ and $\rho$ represent the Lamé constants and density of the corresponding medium.

### 4.2. Computational Model

Based on the experimental model size and parameter design, a three-dimensional numerical model was constructed, as illustrated in Figure 14. Solid elements were used to simulate both the surrounding rock and the secondary lining, while the initial support was represented by shell structure elements [27]. The surrounding rock was modeled using the Mohr–Coulomb yield criterion and an elastic-plastic incremental constitutive relationship, whereas the secondary lining and initial support were characterized by a linear elastic constitutive model. Excavation simulation was carried out utilizing the "model null" function in FLAC3D, with a maximum grid size of 5 m. The model boundary conditions were defined using the free field mesh technology within FLAC3D software to establish free field boundaries, with a viscous boundary applied at the base. The free field boundary comprised four side-layered free field meshes surrounding the model and four corner-point layered free field meshes [28], connected to the main grid via dampers to mimic an infinite field model. The seismic input wave at the base corresponded to the renowned El-centro wave mentioned earlier.

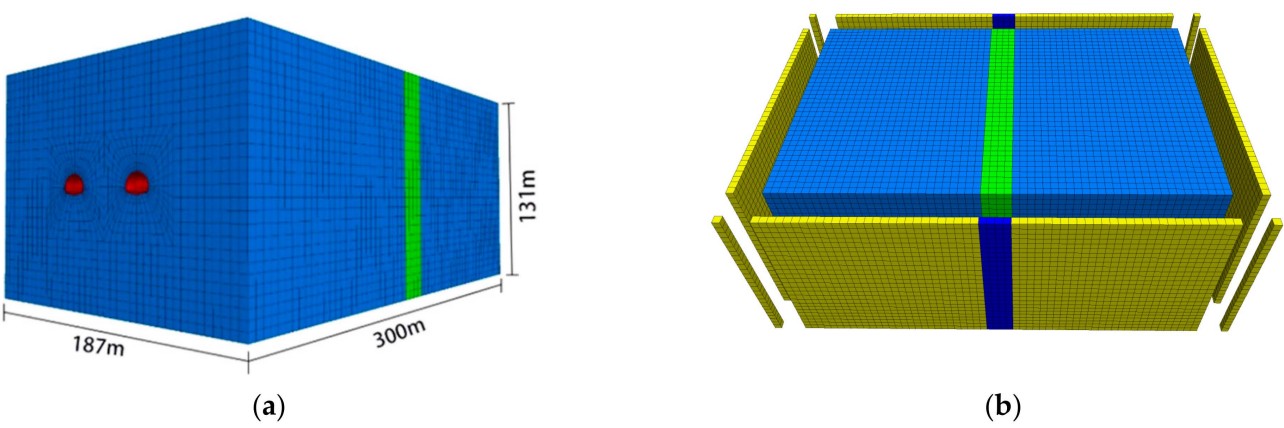

(**a**)  (**b**)

Colour explanation:

Red: tunnel lining; Blue: surrounding rock; Green: fault rock; yellow and deep blue: boundary

**Figure 14.** Diagram of numerical model: (**a**) model grid; (**b**) boundary conditions.

### 4.3. Validation of Numerical Model

Before conducting parametric studies, it was essential to validate the model against shake table test results to ensure the accuracy of the solution and the applicability of the findings to the prototype tunnel design. Accelerations recorded at identical points along tunnel sections under bottom-input El-Centro excitations showed partly agreement between simulation and physical experimentation (Figure 15). This confirmed the numerical model's capability to basically capture soil–structure interaction and seismic response using equivalent FLAC3D techniques.

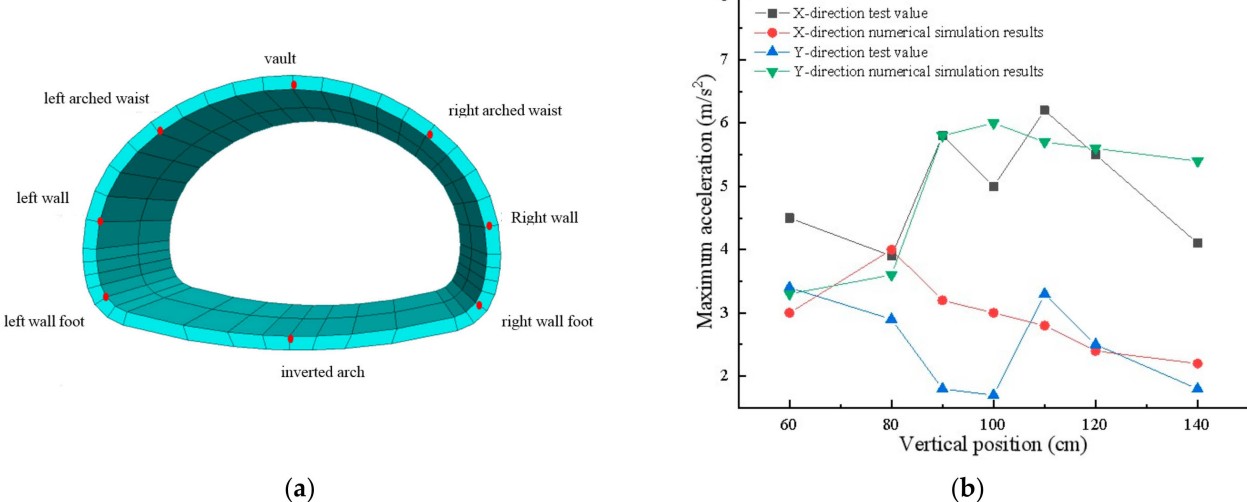

(**a**)                                                                        (**b**)

**Figure 15.** Validation of numerical simulation: (**a**) lining cross section and monitoring points; (**b**) comparison of numerical simulation results with model test results.

### 4.4. Parametric Study of Controls

With validated simulation capabilities, two key anti-vibration measures, namely surrounding rock grouting and shock-absorbing layer insertion, were systematically optimized through numerical experimentation. The evaluation focused on their effectiveness in reducing peak accelerations and stresses in the vicinity of the fault.

#### 4.4.1. Grouting Layer Thickness

In conjunction with current relevant research findings, this section focused on selecting grouting reinforcement surrounding the tunnel rock with grouting layer thicknesses of 2 m, 4 m, and 6 m as seismic fortification measures for the tunnel, and investigated the anti-vibration mechanism of grouting reinforcement surrounding the rock. For the calculation results of peak values of horizontal acceleration (positive and negative) and principal stress at key positions of the tunnel under different grouting thickness conditions, refer to Figures 16 and 17.

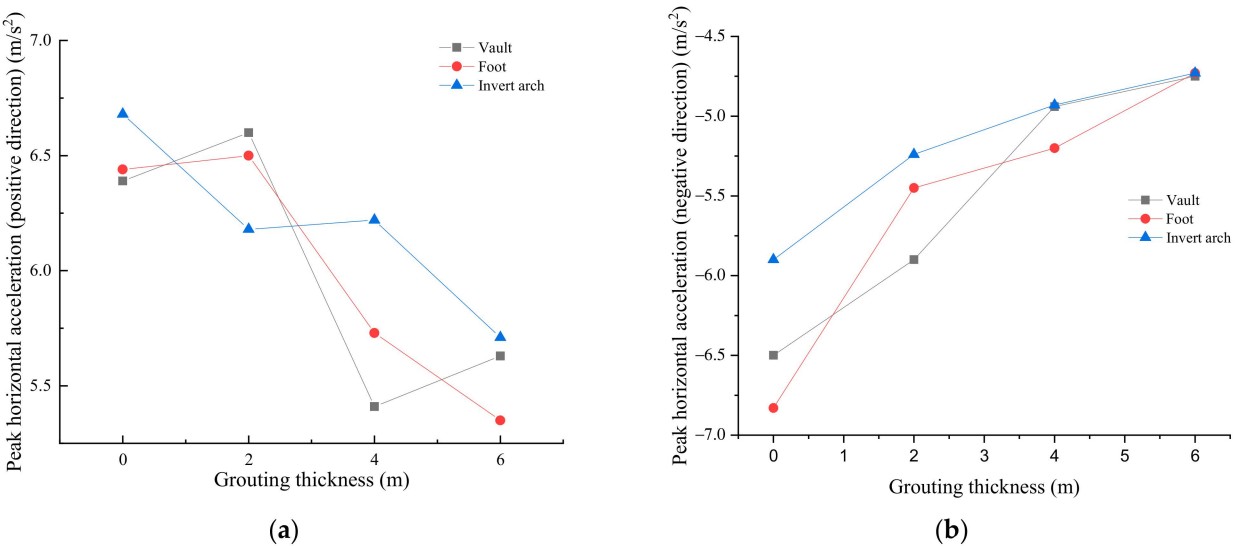

(**a**)                                                                        (**b**)

**Figure 16.** Effect of grouting thickness on horizontal acceleration in (a) positive and (b) negative directions.

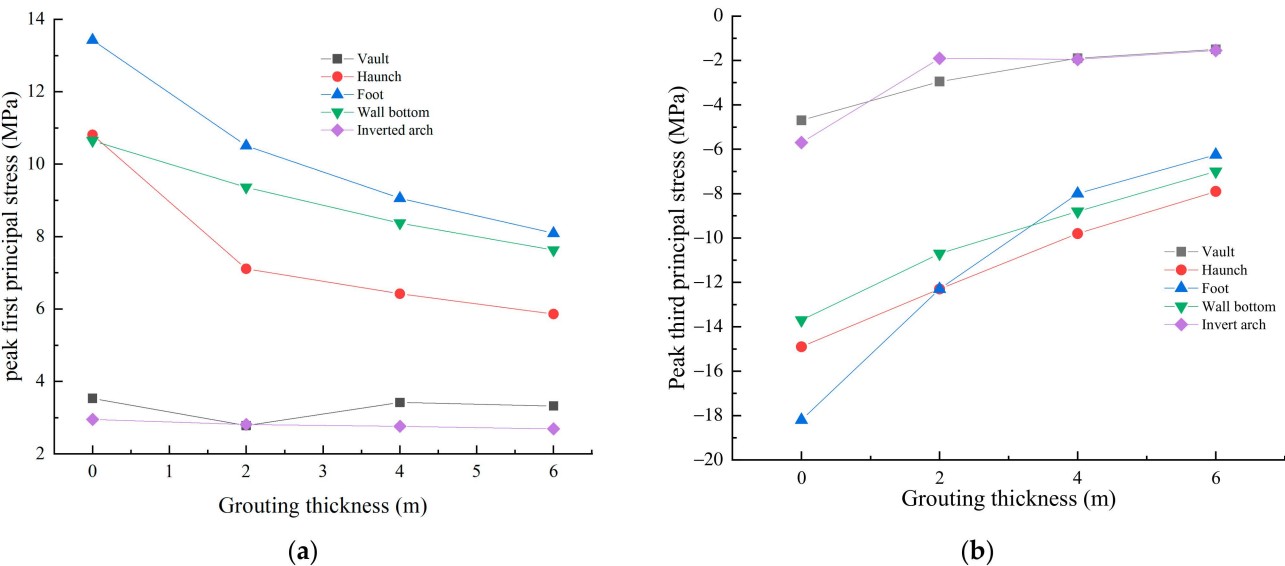

**Figure 17.** Effect of grouting thickness on (**a**) first and (**b**) third principal stresses.

The results in Figure 16 indicated that radial grouting had a similar effect on reducing acceleration at the vault, arch toe, and inverted arch. The maximum acceleration in the fault fracture zone notably decreased with an increase in grouting layer thickness. When the grouting layer thickness was 6 m, the maximum acceleration response peak value of the tunnel lining was only 5.35 m/s$^2$, approximately 17% lower than that without a grouting layer. This was attributed to the significantly enhanced strength of the surrounding rock in the fracture zone after grouting reinforcement, gradually strengthening the constraint of the surrounding rock on the lining structure and reducing the lithological difference between the fracture zone and the intact surrounding rock on both sides.

Furthermore, the acceleration distribution along the longitudinal direction of the tunnel was analyzed. In general, the acceleration response at the fracture zone was larger than that at positions far away from the fracture zone. When the grouting layer thickness was 6 m, the acceleration response peak at the fracture zone position approached that at the intact section of surrounding rock, and the acceleration longitudinal distribution curve became gentler. It could be inferred that when the grouting layer thickness increased to a certain extent, the influence of the fracture zone on the lining essentially disappeared due to the reinforcement of the fractured rock mass around the tunnel over a large range, and the acceleration response curve along the longitudinal direction of the tunnel tended to become horizontal.

In Figure 17, it was evident that the first principal stress and the third principal stress in the tunnel lining structure decreased significantly with the continuous increase in the thickness of the grouting layer at the location of the fault zone and its vicinity. When the thickness of the grouting layer was 2 m, 4 m, and 6 m, respectively, the maximum reduction in the first principal stress was 34%, 41%, and 46%, with the largest reduction occurring at the crown and the base of the wall. The reduction in stress at the crown and invert was not significant. The highest reduction rate of the third principal stress was close to 73%, with the largest reduction occurring at the crown and invert.

Through comparative analysis, it was concluded that the decrease in stress response due to the grouting layer was more significant than the acceleration. This was attributed to the enhanced self-stabilizing ability of the surrounding rock after grouting reinforcement, which increased the constraint force on the tunnel and reduced the external load transmitted from the surrounding rock to the lining under seismic action.

In summary, when the thickness of the grouting layer is 6 m, the principal stress in the tunnel meets the code requirements, the acceleration response of the tunnel along the full length fluctuates within a small range, and the amplification effect of the acceleration

on the lining at the fault zone is already minimal. Further increasing the thickness of the grouting layer no longer has a significant effect. It is suggested that the thickness of the grouting layer should be around 6 m.

4.4.2. Shock Absorbing Layer Thickness

In this section, the shock-absorbing layer was positioned using the "surrounding rock-initial support-shock absorbing layer-secondary lining" method. Four thicknesses were considered: 10 cm, 15 cm, 20 cm, and 30 cm, and compared with the condition without a shock absorbing layer. The peak horizontal accelerations (positive and negative) and peak principal stresses at critical locations in the tunnel for different shock absorbing layer thickness conditions are depicted in Figures 18 and 19.

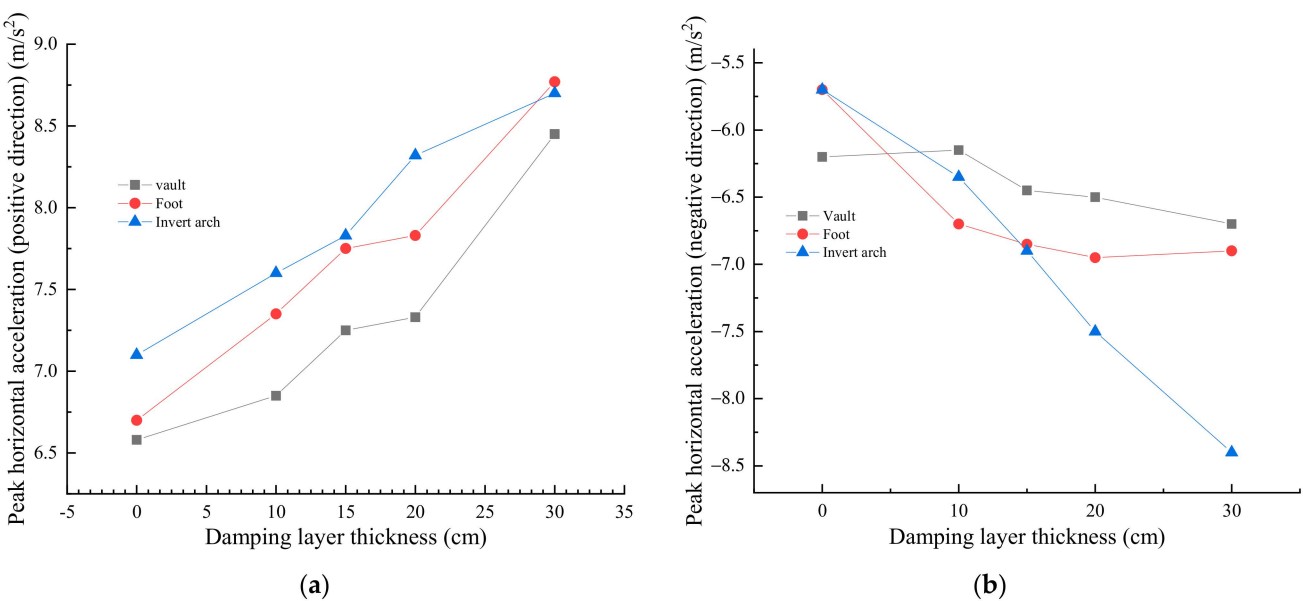

(**a**)                      (**b**)

**Figure 18.** Effect of shock absorbing layer thickness on horizontal acceleration in (**a**) positive and (**b**) negative directions.

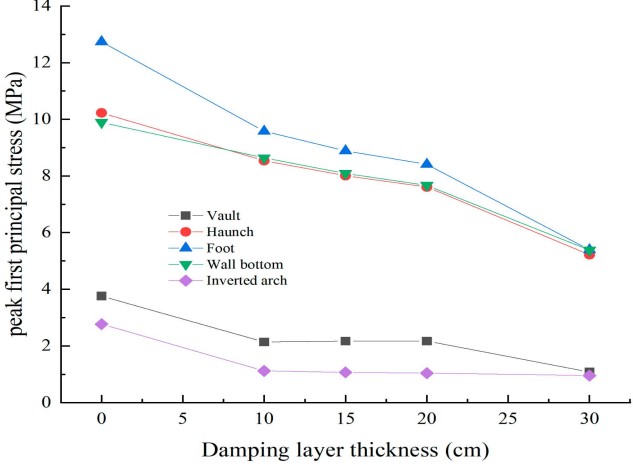

**Figure 19.** Effect of shock absorbing layer thickness on the first principal stress.

The findings from Figure 18 indicated that the amplification factor of the horizontal acceleration at various points of the lining at the fault zone of the extra-large section tunnel aligned with the trend of the change in the thickness of the seismic isolation layer relative to the original seismic wave acceleration. It increased with the thickness of the

seismic isolation layer, and the horizontal acceleration at the crown position was slightly larger than that at the springing and the crown. In the absence of a seismic isolation layer, the maximum acceleration response peak value along the full length of the lining at the fault zone was 6.56 m/s$^2$, approximately 1.89 times larger than the original seismic wave acceleration. With a 10 cm thickness of the seismic isolation layer, the acceleration at the crown increased by 1.97 times, and with a 15 cm thickness, the acceleration at the crown was amplified by 2.09 times. At 20 cm thickness, the amplification factor of the crown acceleration was 2.11 times. The amplification factor of the acceleration exhibited a slowing trend as the thickness of the seismic isolation layer changed from 15 cm to 20 cm. When the thickness of the seismic isolation layer was 30 cm, the amplification factor of the acceleration was 2.52 times, showing an increasing trend again. Upon comparing the acceleration distribution patterns of the fault section and the intact surrounding rock section, it was found that at this point, the difference in horizontal acceleration between the two was the largest.

Combined with related research findings, the initial analysis suggests that the trend of increasing acceleration response with the thickness of the shock absorbing layer can be attributed to the lower and softer elastic modulus of the shock absorbing layer. This leads to reduced restraint on the lining structure, resulting in an amplified acceleration response of the tunnel under seismic action. This observation underscores the need for particular attention in practical engineering anti-seismic efforts.

Figure 19 illustrated that the inclusion of a shock absorbing layer led to a significant reduction in the first principal stress value of the lining structure at the location of the fault fractured zone in the tunnel. Compared to the absence of a shock absorbing layer, the maximum reduction in the first principal stress was nearly 62% after the installation of the shock absorbing layer, with the most pronounced effect observed at the crown and invert positions. The reduction rate slightly increased with the thickness of the shock absorbing layer and gradually stabilized. When the thickness of the shock absorbing layer was below 20 cm, for every 5 cm increase in thickness, the reduction rate consistently remained around 5%. However, once the thickness of the shock absorbing layer exceeded 20 cm, the additional shock absorbing effect became limited. It was recommended that the thickness of the shock absorbing layer should be 20 cm, and potentially increased conservatively if necessary.

### 4.4.3. Overall Analysis

The comparative analysis of horizontal acceleration and maximum reduction amplitude of principal stress of the tunnel under different grouting thickness of surrounding rock and shock absorbing layer thickness conditions is presented in Table 5.

**Table 5.** Analysis of numerical results.

| Parameters | | Grouting Thickness of Surrounding Rock | | | Shock Absorbing Layer Thickness | | | |
|---|---|---|---|---|---|---|---|---|
| | | 2 m | 4 m | 6 m | 10 cm | 15 cm | 20 cm | 30 cm |
| Horizontal acceleration | Maximum reduction ratio/% | −20% | −24% | −31% | 17% | 21% | 23% | 46% |
| | Appearance position | Arch foot | Arch foot | Arch foot | Arch foot | Inverted arch | Arch foot | Inverted arch |
| Principal stress | Maximum reduction ratio/% | −68% | −65% | −73% | −60% | −61% | −62% | −65% |
| | Appearance position | Inverted arch | Inverted arch | Inverted arch | Inverted arch | Inverted arch | Inverted arch | Inverted arch |

The comparative analysis showed that grouting in the surrounding rock significantly reduced the principal stress and acceleration of the tunnel structure, while the installation of a shock absorbing layer notably reduced the stress of the tunnel, albeit with an increase

in acceleration. The existence of a damping layer may lead to a decrease in the impedance of the medium in the outer extension region of the tunnel lining, consequently amplifying the acceleration response. Nonetheless, the stress response of the lining is mitigated due to the diminished deformation modulus in this zone. Both grouting in the surrounding rock and the installation of a shock absorbing layer had a stronger control effect on principal stress than on acceleration, with the maximum reduction amplitudes being 73% and 65%, respectively. The maximum amplitude of principal stress was located at the inverted arch, whereas the maximum amplitude of acceleration was situated at the arch foot and inverted arch. These findings warranted increase attention in practical applications.

Through comparing and analyzing the distribution of peak principal stress and acceleration in tunnel structures under varying operational parameters, it is observed that as the thickness of the damping layer increases, there is a decrease in the peak value of the first principal stress. Meanwhile, the reduction amplitude of the third principal stress peak initially decreases and then increases with thickness. Opting for a 20 cm thick shock absorber layer provides superior shock absorption benefits. A thicker grouting layer results in a greater reduction in peak principal stress. Notably, the seismic resistance effect is more pronounced for grouting thicknesses ranging from 4–6 m, with a recommended thickness of 6 m for optimal application.

## 5. Conclusions

Similar to the conclusions drawn from field investigations and research literature on many historical earthquakes, it was also concludes that most serious seismic damage happens to fault fracture zone [29], and the following conclusions were given based on the above analysis.

(1) Under the excitation of three waveforms, strain, earth pressure, and acceleration all showed a tendency to be slightly higher along the longitudinal direction of the tunnel at the fault section and its surroundings. The most significant amplification effect was observed at the interface between the fault zone and the surrounding rock. This amplification effect extends from the boundary of the surrounding rock to the entire surrounding rock, with a maximum extension distance of 10 m. It is recommended that the minimum length of seismic fortification at both ends of the fault section in tunnels with large spans should be extended by 10 to 20 m beyond that specified for tunnels with the standard section in the Chinese industry standard (JTG D70-2004).

(2) In general, the variation in the anti-vibration effect under different waveform excitations remained largely consistent, affirming the accuracy of the model test results. Following the implementation of seismic joints and shock-absorbing layers, there were changes in the positions of the maximum seismic response peaks at fault interface sections. Grouting reinforcement of the surrounding rock, installation of seismic joints, and incorporation of shock-absorbing layers all demonstrated noticeable seismic effects. The seismic impact of seismic joints and shock-absorbing layers was comparable, whereas the reinforcement effect of grouting on the surrounding rock surpassed that of the other two measures.

(3) As the thickness of the grouting layer increased, there was a significant decrease in both the acceleration and the peak value of principal stress in the tunnel structure. The influence of the grouting layer on stress response was more pronounced than on acceleration, with the maximum reduction in principal stress reaching up to 73%. When the grouting layer thickness reached 6 m, the tunnel principal stress complied with the code requirements. At this thickness, the acceleration response peak value at the fracture zone position approached that of the intact section of the surrounding rock, the longitudinal distribution curve of acceleration became gentler, the amplification effect of lining acceleration at the fracture zone position diminished significantly, and the impact of further increasing the grouting layer thickness became negligible. It was recommended that the grouting layer thickness be maintained at approximately 6 m.

(4) The horizontal acceleration of the lining at the fault of an extra-large section tunnel increased with the thickness of the shock-absorbing layer. When the shock-absorbing layer thickness was 30 cm, the acceleration amplification reached 2.52 times, representing the largest difference. The installation of a shock-absorbing layer could significantly reduce the first principal stress value of the lining structure at the fault fracture zone, with the maximum reduction rate nearing 62%. Initially, the reduction rate slightly increased with the thickness of the shock-absorbing layer and gradually stabilized. Once the shock-absorbing layer thickness exceeded 20 cm, the effectiveness of further increasing the thickness became limited. It was recommended to maintain the shock-absorbing layer thickness at 20 cm.

**Author Contributions:** Conceptualization, F.Z. and B.L.; methodology, B.L. and N.Z.; software, F.Z. and B.L.; validation, F.Z., B.L. and N.Z.; formal analysis, F.Z. and B.J.; investigation, F.Z. and B.L.; resources, F.Z. and B.L.; data curation, F.Z. and B.J.; writing—original draft preparation, F.Z. and N.Z.; writing—review and editing, F.Z., B.L. and B.J.; visualization, B.L.; supervision, B.L.; project administration, B.L. All authors have read and agreed to the published version of the manuscript.

**Funding:** This research was funded by the Science and Technology Research Project of Chongqing Education Commission, grant number (with grant number KJQN202205802, KJQN202303436, KJQN202305802, and KJQN202304301), Construction Science and Technology Plan Project of Chongqing (with number Chengke Zi 2023 No. 1-10) and General Program of Natural Science Foundation of Chongqing (with number CSTB2022NSCQ-MSX0661).

**Institutional Review Board Statement:** Not applicable.

**Informed Consent Statement:** Not applicable.

**Data Availability Statement:** Data will be made available on request.

**Conflicts of Interest:** The authors declare no conflicts of interest.

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
