# Peer review of "Shaking Table Testing and Numerical Study on Aseismic Measures of Twin-Tube Tunnel Crossing Fault Zone with Extra-Large Section"

_applsci, doi:10.3390/app14062391_

Round 1

Reviewer 1 Report

Comments and Suggestions for Authors

The authors of the manuscript submitted for review presented a very interesting study on the dynamic response of a tunnel passing through a tectonic fault. The study considered the influence of three different seismic control measures: surrounding rock grouting, seismic joints, and shock absorption layers, on the acceleration and stress values within the structure. The paper included both scientifically and practically significant engineering research conducted on a shaking table and extensive numerical analyses of the problem.

The article is clear and well-thought-out, and the conclusions drawn by the authors are correct. However, there are several elements in the paper that need to be improved before the manuscript can be published. The following issues require corrections or commentary:

General comments:

1.      The authors conduct model studies on a shaking table. The constructed model represents a complex structure in terms of materials. However, the authors do not refer to any theory related to similarity criteria. They do not present any criteria or similarity scales. The authors do not mention, for example, whether the similarity criterion for the elasticity of the surrounding ground and the material of the tunnel is identical or different. If different, how did they interpret the real results? In the case where the ratio Emodel/Eprototype is different for the tunnel material than for the ground, the relative stress ratio obtained for the tunnel and ground during the study will be different from reality.

They have not mentioned which parameters formed the dimensional basis and what criteria they depended on. According to the reviewer, this requires detailed explanations and supplementation of the literature, especially since the authors refer in the article to some similarity scales (e.g., time and acceleration scale).

2.      In Section 4.3 of the paper, the authors state that the results obtained from the shaking table experiment and numerically are in "close agreement." However, such a conclusion seems contradictory considering that in Figure 15, the accelerations in the X direction (and also in the Y direction) obtained from both methods are completely divergent. Could there have been a mistake in labeling the X and Y directions? Please provide clarification.

3.      The authors of the paper have shown that for rock grouting causing a decrease in acceleration amplitudes, a decrease in stresses is observed. A similar decrease in stresses was observed for damping layers, even though the application of this solution leads to an increase in acceleration amplitudes. According to the authors, what is the mechanism behind this phenomenon? Please provide a comment.

Editorial comments:

1.      Please consider enlarging Figure 3. Images of the experimental model and the measurement setup are always interesting due to the authors' contribution to the work.

2.      The formatting of the references does not meet the journal's criteria. The citation style is inconsistent; for example, sometimes all authors of the cited publication are listed (see [5]), while at other times the abbreviation "et al." is used (see [6]).

Overall recommendation: Accept after Major Revisions

Reviewer 2 Report

Comments and Suggestions for Authors

Refer to the attached PDF file.

Comments on the Quality of English Language

Unfortunately, there are still some grammatical errors and instances of badly written sentences. Therefore, the authors should revise the manuscript and refine the language carefully again.

Reviewer 3 Report

Comments and Suggestions for Authors

To enhance the paper, consider expanding on the following areas:

1- Provide a more detailed description of the geological settings, including the types of rocks, soil profiles, and the specifics of the fault zones. This could include the geological history, stratigraphy, and any previous seismic activity in the area. Incorporating detailed geological maps and cross-sections can also improve understanding.

2- A more thorough analysis of the fracture network within the fault zone could be beneficial. This includes the orientation, density, and connectivity of fractures and how they influence seismic wave propagation and tunnel deformation. Discussing the role of the fracture network in the local amplification of seismic waves or in directing seismic energy could offer deeper insights.

3- Although the study might already include material properties used in the models, expanding on how these properties were determined (e.g., through laboratory tests or literature) and their variability could strengthen the study's credibility. Discussing the implications of material heterogeneity on the model's outcomes could also be insightful.

4- Elaborate on the selection and scaling of seismic input motions for both the shake table tests and numerical models. Discussing the rationale behind choosing specific earthquake records (e.g., El-Centro, Taft, and artificial synthetic waves) and how well they represent the seismic risk of the tunnel location could improve the paper's comprehensiveness.

5- Including a sensitivity analysis on key parameters such as the thickness of the grouted zones, properties of the shock absorption layers, and the spacing of seismic joints could offer valuable insights into the robustness of the seismic mitigation measures.

6- If possible, comparing the study's findings with data from actual earthquakes affecting similar tunnels could greatly enhance the paper's relevance and applicability.

Future Research Directions: Concluding with specific recommendations for future research, such as exploring alternative aseismic measures, conducting field tests, or developing more sophisticated numerical models, could make the paper a more valuable resource for the engineering community.

Comments on the Quality of English Language

Moderate editing of English language required

Reviewer 4 Report

Comments and Suggestions for Authors

I recommend the publication of this article after MAJOR changes.

Article overview:

The main question addressed by the research is the determination of the appropriate anti-vibration measures that mitigate longitudinal fault encountered during the construction of the Xiangyangshan Highway tunnel.

The authors studied the realistic 1:100 scaled model on a shaking table with El-Centro, Thaft, and synthetic wave instigation and verified the results with the finite difference program FLAC3D. Using experimental and numerical analysis, they concluded that 6 m thick grouted zones and 20 cm thick rubber interlayers between tunnel lining shells maximize structural reinforcement against fault zone effects.

However, this topic is not original, and the relevance of this article is limited because it does not showcase any significant improvement compared to the cited state-of-the art papers, nor does it fill a significant gap in the current literature.

The main issue that the authors need to fix before this article becomes worthy of publication is the literature overview in the introduction. They need to emphasize the drawbacks and shortcomings of current procedures and compare their work with cited papers, clearly identifying the improvements and benefits of their methodology.

Lines 133 to 139 are completely redundant; an image of the Xiangyangshan tunnel would be much more appropriate.

Line 147 is missing a reference for rock quality grading, as is line 149.

The source of earthquake waves should be cited in line 154.

Round 2

Reviewer 1 Report

Comments and Suggestions for Authors

The authors addressed all the comments included in the review.

However, the adopted similarity scales are still not sufficiently explained.

However, the manuscript is suitable for publication.

Author Response

Response to Reviewer 1 Comments:

Thank you again very much for taking the time to review the revised manuscript. 

We have further improved the language of the paper.

Reviewer 3 Report

Comments and Suggestions for Authors

Please increase the size of the legend in your figures.

Comments on the Quality of English Language

Minor revision in editing.

Author Response

Response to Reviewer 3 Comments:

Thank you again very much for taking the time to review the revised manuscript. 

We have further improved the language of the paper and adjusted the size of the legend in the figures.

Reviewer 4 Report

Comments and Suggestions for Authors

Authors fixed discovered issues and greatly improved their manuscript, which is now ready for publication

Author Response

Response to Reviewer 4 Comments:

Thank you again very much for taking the time to review the revised manuscript. 

We have further improved the language of the paper.